# Effects of Acute Aerobic Exercise Combined with Resistance Exercise on Neurocognitive Performance in Obese Women

**DOI:** 10.3390/brainsci10110767

**Published:** 2020-10-22

**Authors:** Huei-Jhen Wen, Chia-Liang Tsai

**Affiliations:** 1Physical Education Center, College of Education and Communication, Tzu Chi University, Hualien 97004, Taiwan; 2Sports Medicine Center, Tzu Chi Hospital, Hualien 97004, Taiwan; 3Institution of Physical Education, Health and Leisure Studies, National Cheng Kung University, Tainan 70101, Taiwan

**Keywords:** obesity, inhibitory control, event-related potential, aerobic exercise, resistance exercise

## Abstract

To the best of the author’s knowledge, there have been no previous studies conducted on the effects of a combination of acute aerobic and resistance exercise on deficit of inhibitory control in obese individuals. The aim of this study was, thus, to examine the effect of a single bout of such an exercise mode on behavioral and cognitive electrophysiological performance involving cognitive interference inhibition in obese women. After the estimated VO_2_max and percentage fat (measured with dual-energy X-ray absorptiometry (Hologic, Bedford, MA, USA) were assessed, 32 sedentary obese female adults were randomly assigned to an exercise group (EG) and a control group (CG), with their behavioral performance being recorded with concomitant electrophysiological signals when performing a Stroop task. Then, the EG engaged in 30 min of moderate-intensity aerobic exercise combined with resistance exercise, and the CG rested for a similar duration of time without engaging in any type of exercise. After the interventions, the neurocognitive performance was measured again in the two groups. The results revealed that although acute exercise did not enhance the behavioral indices (e.g., accuracy rates (ARs) and reaction times (RTs)), cognitive electrophysiological signals were improved (e.g., shorter N2 and P3 latencies, smaller N2 amplitudes, and greater P3 amplitudes) in the Stroop task after the exercise intervention in the EG. The findings indicated that a combination of acute moderate-intensity aerobic and resistance exercise may improve the neurophysiological inhibitory control performance of obese women.

## 1. Introduction

Obesity is considered to be an immunodeficient, chronic inflammation state, which may contribute to an increased risk of premature death [1]. This chronic disease has been associated not only with increases in non-communicative diseases (e.g., type II diabetes, hypertension), but also with reduced brain volume (e.g., frontal cortex and anterior cingulate cortex) and impaired neurocognitive outcomes (e.g., frontal-lobe-based executive functions) [2]. In particular, attentional inhibition and inhibitory control are cognitive domains that are affected negatively by obesity [3,4,5], reflecting a deficit in the neural networks within the anterior cingulate or prefrontal cortex [6,7].

Acute exercise is defined as a single bout of exercise lasting from a few seconds to as long as several hours [8]. Acute exercise can enhance a wide range of cognitive performance, including basic information processing, attention, crystallized intelligence, and executive functions [9,10]. Acute exercise has also been shown to be associated with improvements in the inhibition process [11], a subcomponent of executive functions modulated by the dorsolateral prefrontal cortex, which is known to be particularly affected by exercise [12] in healthy children and adults [13,14].

In addition, a comparison of lean individuals with obese individuals indicated that acute exercise leads to decreased neural responses to food cues compared with non-food cues, suggesting different effects of exercise on the neural processing of food cues based on weight status [15]. Quintero et al. (2018) also found that, compared to an acute bout of progressive resistance exercise (PRE), both acute high-intensity aerobic interval exercise (HIIE) and PRE + HIIE interventions significantly enhance behavioral performance on cognitive inhibition and attention capacity in overweight inactive male adults when performing the Stroop test and d2 test of attention [16]. However, by contrast, Tomporowski et al. (2008) reported that a 23-min bout of treadmill walking did not influence error rates and global switch cost scores in overweight children when performing a task-switching task [17]. Also, one study found no reliable improvements in executive functions (e.g., Stroop and Go/nogo tasks) after a 30-min bout of moderate-intensity aerobic exercise among adults with co-morbid overweight/obesity and type 2 diabetes [18]. However, female participants and those who were more physically active showed reduced Stroop interference scores following moderate exercise [18]. Accordingly, the effects of acute exercise on cognitive performance are still inconsistent in overweight/obese event-related potential (ERP), and age- and gender-related differences may exist in this population [16,17,18]. Additionally, although a previous study reported the beneficial effects of acute aerobic exercise on behavioral cognitive control in overweight inactive male adults [16], whether acute exercise affects behavior and, especially, neurophysiological (e.g., ERP) signals involving attentional and inhibitory control in obese female adults is still worthy of further investigation.

Electroencephalographic signals can provide insights into the effects of exercise on cognition [19,20]. The ERP components of the brain have been demonstrated to be selective in terms of differentiating cognitive electrophysiological performance in obese and normal-weight individuals [5]. Previous studies have reported deviant behavioral (e.g., slower reaction times (RTs)) and ERP performance (e.g., longer N2 and P3 latencies and smaller P3 amplitudes) in obese individuals when they are performing various cognitive tasks involving attentional inhibition and inhibitory control [3,4,5]. Since the Stroop task involves a complex cognitive interference inhibition process, and female adults suffering from obesity show deficits in such underlying neural systems [5], this type of cognitive task was used to assess the effects of an acute exercise intervention on inhibitory control [5,19] in obese female adults in the present study.

According to the American College of Sports Medicine (ACSM), appropriate physical activity intervention strategies for weight loss are recommended for a minimum of 150 min per week comprising moderate-intensity aerobic exercise combined with resistance exercise for overweight and obese adults to improve their health [21]. Accumulating evidence supports the positive influence of acute aerobic exercise on attentional control/inhibitory control in healthy individuals [12,13,14,22,23]. In addition, some studies have proven that neurocognitive performance can be enhanced via acute and chronic resistance exercise [10,24,25,26]. A combination of aerobic and resistance exercise has also been proven to be beneficial to cognitive functions in patients with stroke [27] and dementia [28], as well as in overweight inactive adult men [16]. This type of exercise mode has even been found to have stronger effects on executive functions (i.e., inhibition, impulse control, planning, and set-shifting) in in adolescents [29], in healthy sedentary adults [30], and older adults [31] as compared to an aerobic exercise program alone since a combined aerobic and resistance exercise program can simultaneously positively increase the levels of brain-derived neurotrophic factors [14,23,25,32] and insulin-like growth factor-1 [10,25,26,32]. In addition, resistance exercise may improve cognitive functions specifically through lowering the levels of neurotoxic homocysteine [26,31]. Therefore, moderate-intensity aerobic exercise combined with resistance exercise may be a potential effective exercise mode, as suggested by the ACSM that can be used to improve neurocognitive problems related to inhibitory control in obese individuals.

Obesity is caused by a number of multidimensional factors [33] and results in deficits in neural circuits related to inhibitory control [4,7,22,34]. Obese adults have been shown to obtain advantages related to neurocognitive performance through engaging in regular exercise as compared to obese individuals with a sedentary lifestyle [35], suggesting that obesity does not preclude benefits derived from physical exercise and cardiorespiratory fitness to cognitive functions and neural networks. Although a number of studies have reported that a combination of aerobic and resistance exercise mode may effectively improve cognitive behavioral performance in overweight inactive male adults [16] and in patients with stroke [27] and dementia [28], thus far, a paucity of data exist on the possible additive effects of an acute bout combining these two exercise modes on behavioral and cognitive electrophysiological performance involving inhibitory control in obese female individuals. Therefore, the purpose of the current study was to investigate the potential effects of acute aerobic exercise coupled with resistance exercise on neurocognitive performance related to cognitive interference inhibition in sedentary obese women when performing the Stroop task. Based on the previous findings mentioned above, we hypothesized that an acute bout of a program combining aerobic exercise and resistance exercise is a feasible and effective intervention that can improve cognitive deficits in obese female individuals.

## 2. Materials and Methods

### 2.1. Ethical Approval

This research meets the standards set by terms of the Declaration of Helsinki. All protocols in this study were approved by the Research Ethics Committee at Hualien Tzu Chi Hospital, Hualien, Taiwan (IRB105-61-A). The participants were freely to withdraw their consent at any time during the course of the study without any reason.

### 2.2. Participants

An experiment procedure of the Consolidated Standards of Reporting Trials (CONSORT) outlining the number of participants for the present study is shown in Figure 1. This study was broadcasted on a local radio station and also advertised in a local newspaper in Hualien County, Taiwan. Based on the criteria for obesity established by the Western Pacific Regional Office of the World Health Organization for Asian populations according to related mortality and morbidity risks [36,37], females with body mass index (BMI) > 25.0 kg/m^2^ were recruited as participants in the present study. Additional inclusion standards were (a) right-handed, as assessed by the Edinburgh Handedness Inventory using the arbitrary cut-off points between 0 to ±60 [38,39], (b) non-smokers, (c) normal or corrected-to-normal vision, (d) no symptoms of depression as measured by the Beck depression inventory II (BDI-II; all scored below 13) [40], (e) cognitive integrity measured by the Mini-Mental State Examination (MMSE; all scored above 24) [41].

According to these criteria, and after determining a priori power for a repeated measures analysis of variance (RM-ANOVA) using G-Power 3.1, the minimum sample size of ~16 participants for each group was required to reach a power of 80% and moderate effect sizes [42]. Fifty women from the community in Hualien City were interested in participating in the study. Eighteen were excluded because of not meeting the criteria or declined to participate after hearing a detailed explanation of the protocol. Eligible participants were self-reported to be free of metabolic or cardiovascular diseases, neurological, or psychiatric disorders and professed to be free of a history of brain injury, or medication intake that would influence central nervous system (CNS) functioning. Thirty-two eligible healthy women with obesity were then randomized to an exercise group (EG) or a control group (CG). The demographic characteristics data for the two groups are provided in Table 1.

### 2.3. Experimental Procedure

All participants visited the study site twice. At the first visit, dual-energy X-ray absorptiometry (DXA) and physical fitness tests (2 km walk test and submaximal leg extension strength test) were scheduled to respectively assess the body composition and cardiovascular/muscular fitness of the participants at Tzu Chi Hospital (Hualien, Taiwan) and the cognitive neurophysiology laboratory at Tzu Chi University (Hualien, Taiwan).

Each participant had a 2nd visit at approximately 7:50–08:30 AM in the same week. All participants were asked to refrain from caffeine or alcohol intake and strenuous exercise for 24 h. The research assistant explained the experimental procedure to each participant and then asked her to complete an informed consent form, a medical history, a handedness inventory, a demographic questionnaire, the MMSE, and the BDI-II. After completing all of the questionnaires, each participant sat comfortably 80 cm in front of a laptop screen in a semi-dark room. An electrocap and electro-oculographic (EOG) electrodes were then attached to the scalp and face of each participant prior to the cognitive tests. After some practice trials, a simultaneous formal cognitive task test with concomitant electrophysiological recording was performed. Then, the participants in the EG performed a single 30-min bout comprising a combination of moderate-intensity physical exercises (please see the detailed protocols in Section 2.4.). After engaging in acute exercise and after the participants’ heart rate (HR) had returned to within 10% of pre-exercise levels (mean 28.4 ± 10.8 min), they completed the cognitive task along with ERP recording again. In the CG, the participants were instructed to sit quietly for 30 min, after which they took the cognitive task test again.

### 2.4. Exercise Intervention

The participants in the EG were taught to determine their target exercise HR, and HR was monitored throughout the exercise session using a telemetry HR monitor (S810, Polar, Kempele, Finland).
Target exercise HR = [(220−age)−HR rest] × 50% + HR rest


The exercise was held in the laboratory. The exercise program (see Figure 2) consisted of a 3–5 min warm-up session, 30-min of supervised moderate-intensity aerobic dance (i.e., corresponding to 55% of the individual target HR reserve (HRR) alternately combined with dumbbell resistance exercises specifically designed for this study, followed by a 3–5 min cool-down session.

The exercises involved 4 cycles accompanied with music at 126 beats per min. The next cycle was started after a break between cycles when the HR reached the 50% HRR of the individual’s target HR. There were 6 sets of aerobic dance and resistance exercises alternating in each cycle in 1 min sets following a 15 s break to drink water and stretch. Approximately 12–16 repetitions per set of six dumbbell/bodyweight resistance exercises targeting the major muscle groups were carried out at moderate intensity [43]. The repetition velocity of each resistance training movement was set to: shoulder extension with arm pronation (8.4 s of slow motion to 2.1 s of normal pace), arm curl (4.2 s of slow motion to 2.1 s of normal pace), and elbow extension (8.4 of slow motion to 2.1 s seconds of normal pace). One set in each session was supervised and led by a trained instructor. The instructor led participants through a full range of motion for each movement, which expended between 350 and 500 kcal according to the participant’s weight. The investigator provided verbal encouragement throughout the exercise period. The average intensity of all participants in the EG was 60.05 ± 3.57% HRR when they performed the aerobic exercise.

### 2.5. Whole and Regional Body Composition

Body composition was measured using DXA (Discovery Wi, Hologic Inc., Bedford, MA, USA). The measurement was performed by a certified technician according to the standard operating procedure. The scanning instructions and procedures were standardized for all participants. The trunk region included the area from the bottom of the neckline to the top of the pelvis, excluding the arms. The mass output from the DXA scanner was expressed in grams. Each testing day, the accuracy of the densitometer was calibrated using the manufacturer’s spine phantom with a known hydroxyapatite density.

### 2.6. Cognitive Task- Stroop Task

A two-choice Stroop task inducing inhibitory control effects of executive functions in both young and older adults [44] programmed using E-prime (Psychology Software Tools, Sharpsburg, PA, USA) was adopted in the current study. Because semantics interferes significantly with the naming of colors [45], and color interferes very little with reading words [46,47], the color-naming condition was used to investigate the effect of acute exercise on neurocognitive functioning associated with cognitive interference inhibition in obese women. The stimuli, two color names in Chinese presented as 4.5 × 4.5 cm letters in “紅” (red) and “綠” (green), were displayed in the center of a 21-in. cathode-ray tube against a black background at viewing distance of 80-cm. In the congruent condition, the meaning of the word matched its color, whereas the color of the word was different from its word meaning in the congruent condition. A single test block consisted of an equal number of both incongruent and congruent trials in a randomized order. A total of 240 trials were divided into two blocks of 120 trials, with a rest period of 2 min between blocks. Each stimulus appeared on the screen until the participant responded, and the next stimulus appeared 1.5 to 2 s after the response. The participants were asked to press the computer keyboard as accurately and quickly as possible in response to the color while ignoring the word meaning. The stimulus response pairs were counterbalanced across participants. All participants performed the Stroop task along with simultaneous cognitive electrophysiological recording. After a practice block of 10 trials to ensure that the participants understood the task rules, the formal test was administered to allow for data collection of neurocognitive performance.

### 2.7. Event-Related Potential (ERP) Recording and Analysis

Brain electrical activity was recorded using the eegoTM amplifier system (ANT Neuro, EE-211, revision Nr 1.2, Berlin, Germany) from 64 scalp sites (10–10 system) with Ag/AgCl electrodes mounted in an elastic cap. The raw electroencephalography (EEG) signal was acquired at an A/D rate of 500 Hz/channel using a 60-Hz notch filter and a 0.1–50 Hz band-pass filter. All inter-electrode impedance was kept below 5 KΩ. Prior to averaging the ERP components, an offline electrooculographic correction was applied to the individual trials. All trials with artefacts (i.e., electromyogram and electrooculogram readings exceeding ± 100 µV) and response errors were eliminated. The remaining effective ERP data in the Stroop task were separately averaged offline and constructed from congruent and incongruent conditions over a 1000 ms epoch beginning 200 ms before the onset of the target stimulus. The mean latencies and amplitudes of the P2, N2, and P3 components were measured for the Fz and Cz electrodes [48]. The time windows for detection of the P2, N2, and P3 components were 150–275 ms, 200–400 ms [49], and 300–800 ms [50,51], respectively. Latency was defined as the time at which the peak amplitude was reached within the latency window for every participant and was calculated as the time in milliseconds.

### 2.8. Data Processing and Statistical Analyses

The behavioral performance in the Stroop task was assessed with the accuracy rates (ARs) (%) and RTs (milliseconds) to each target presentation automatically calculated by the E-prime software (Psychology software tool, Pittsburgh, PA, USA). Responses later than 1500 ms and sooner than 200 ms after target onset were excluded in both the congruent and incongruent conditions.

Descriptive statistics are presented for the basic demographic characteristics of the participants. For the behavioral analyses (i.e., ARs and RTs) and cognitive electrophysiological (i.e., P2, N2, and P3 latencies and amplitudes), all of the independent variables were analyzed with a two-way repeated measures analysis of variance (RM-ANOVA) (i.e., ***Group*** (EG vs. CG) × ***Time*** (pre- vs. post-test) × ***Condition*** (congruent vs. incongruent) × ***Electrode*** (Fz vs. Cz)). If the assumption of sphericity was violated, analyses employing the Greenhouse–Geisser correction with three or more within-subject levels were conducted. Where a significant difference occurred, Bonferroni *post-hoc* analyses were performed. Partial Eta squared (*η*_p_^2^) was used to calculate the effect sizes of significant main effects and interactions, with the following standard used to determine the magnitude: <0.08 (small effect size), between 0.08 and 0.14 (medium effect size), and >0.14 (large effect size). A *p*-value < 0.05 was accepted as statistically significant.

## 3. Results

### 3.1. Demographic Data

As shown in Table 1, there were no significant between-group differences in the weight and circumference measures (e.g., BMI, waist girth, abdominal girth, and hip girth), body composition status, estimated VO_2_max, and blood pressure (e.g., systolic blood pressure (SBP) and diastolic blood pressure (DBP)). In addition, no significant differences were found in the values for the other demographic measures.

### 3.2. Behavioral Performance of Stroop Task

The pre- and post-test behavioral performance of the EG and CG groups are shown in Figure 3.

• Accuracy rate (AR)

The RM-ANOVA for the ARs showed significant main effects of ***Group*** [F_(1,__30)_ = 7.17, *p* = 0.013, *η*_p_^2^ = 0.22] and ***Condition*** [F_(1,__30)_ = 57.38, *p* < 0.001, *η*_p_^2^ = 0.69], with the ARs in the EG (98.06 ± 1.45%) being significantly higher than those in the CG (95.31 ± 3.78%) across two time points and two electrodes, and with the ARs being significantly higher in the congruent condition (98.70 ± 2.04%) as compared to those in the incongruent condition (95.07 ± 4.18%). Neither a significant main effect of ***Time***, nor significant interactions between ***Time***, ***Group***, and ***Condition*** were found.

• Reaction time (RT)

The RM-ANOVA for the RTs showed significant main effects of ***Time*** [F_(__1,30)_ = 12.30, *p* = 0.002, η_p_^2^ = 0.31] and ***Condition*** [F_(1,30)_ = 40.80, *p* < 0.001, η_p_^2^ = 0.59], with the RTs being significantly faster in the post-test (543.04 ± 72.49 ms) than in the pre-test (571.40 ± 80.90 ms) across both groups and two conditions, with the RTs being significantly faster in the congruent condition (530.05 ± 56.38 ms) as compared to in the incongruent condition (584.40 ± 93.72 ms). Neither a significant main effect of ***Group*** nor significant interactions between ***Time***, ***Group***, and ***Condition*** were found.

### 3.3. Electrophysiological Performance

Figure 4 displays the grand-average ERP waveforms for the two midline electrodes pre- and post-test in the two groups when performing the Stroop task.

• P2 component

The RM-ANOVA for the P2 latency revealed neither main effects of ***Time*** and ***Group*** nor significant interactions among ***Group***, ***Time***, ***Condition***, ***or***
***Electrode***.

The RM-ANOVA for the P2 amplitude showed that there was a significant main effect of ***Time*** [F_(__1,__30)_ = 47.71, *p* < 0.001, *η_p_*^2^ = 0.61], with the P2 amplitude being significantly greater in the post-test (3.91 ± 0.68 *μ*V) than in the pre-test (2.97 ± 0.52 *μ*V) across both groups for two conditions and two electrodes. No significant interactions of ***Group***, ***Time***, ***Condition***, or ***Electrode*** were found.

• N2 component

The RM-ANOVA for the N2 latency showed significant main effects of ***Time*** [F_(1,30)_ = 8.26, *p* = 0.007, *η*_p_^2^ = 0.22] and ***Condition*** [F_(1,30)_ = 14.94, *p* = 0.001, *η*_p_^2^ = 0.33], where the N2 latency in post-test (260.09 ± 68.69 ms) was shorter than in the pre-test (269.78 ± 72.69 ms) across both groups for two conditions and two electrodes, and where the N2 latency in the congruent condition (264.08 ± 69.23 ms) was shorter than that in the incongruent condition (265.80 ± 69.86 ms) across both groups, two time points, and two electrodes. These main effects were superseded by significant interactions of ***Group*** × ***Time*** [F_(1,30)_ = 26.65, *p* < 0.001, *η_p_*^2^ = 0.47] and ***Group*** × ***Time*** × ***Condition*** [F_(1,30)_ = 7.19, *p* = 0.012, *η_p_*^2^ = 0.19]. The *post-hoc* analyses revealed that, when compared to the pre-test, the N2 latency in the EG was significantly shorter post-test in both the congruent (pre- vs. post-test: 277.19 ± 78.97 ms vs. 249.25 ± 64.51 ms, *p* < 0.001) and incongruent (pre- vs. post-test: 278.84 ± 79.36 ms vs. 253.94 ± 67.47 ms, *p* < 0.001) conditions. In contrast, the N2 latency was significantly longer post-test relative to pre-test in the CG in the congruent condition (pre- vs. post-test: 260.25 ± 66.86 ms vs. 268.13 ± 72.45 ms, *p* = 0.025).

In terms of N2 amplitude, there was a significant main effect of ***Condition*** [F_(1,30)_ = 33.14, *p* < 0.001, *η*_p_^2^ = 0.53], with the N2 amplitude being significantly smaller in the incongruent condition (−0.94 ± 0.23 *μ*V) as compared to in the congruent condition (−0.72 ± 0.16 *μ*V) across two time points, two groups, and two electrodes. These main effects were superseded by significant interactions of ***Group*** × ***Time*** [F_(1,30)_ = 96.18, *p* < 0.001, *η*_p_^2^ = 0.76] and ***Group*** × ***Time*** × ***Condition*** [F_(1,30)_ = 5.92, *p* = 0.021, *η*_p_^2^ = 0.17]. The *post-hoc* analyses revealed that, when compared to the pre-test, the N2 amplitude in the EG was significantly smaller post-test in both the congruent (pre- vs. post-test: −0.72 ± 0.16 *μ*V vs. −0.52 ± 0.09 *μ*V, *p* < 0.001) and incongruent (pre- vs. post-test: −1.26 ± 0.26 μV vs. −0.90 ± 0.25 μV, *p* < 0.001) conditions. No change was found in the CG for either condition pre- and post-test.

• P3 component

The RM-ANOVA for the P3 latency revealed a significant main effect of ***Condition*** [F_(1,30)_ = 9.23, *p* = 0.005, *η_p_*^2^ = 0.24], with the P3 latency being significantly shorter in the congruent condition (450.39 ± 133.59 ms) as compared to in the incongruent condition (463.67 ± 133.45 ms, *p* = 0.019) across the two time points, the two groups, and two electrodes. The interaction of ***Group*** × ***Time*** [F_(1,30)_ = 8.55, *p* = 0.007, *η_p_*^2^ = 0.22] was also significant. The *post-hoc* analyses revealed that the P3 latency was significantly shorter post-test compared to in the pre-test only in the EG (pre- vs. post-test: 445.31 ± 128.27 ms vs. 421.88 ± 126.59 ms, *p* = 0.002) across two conditions and two electrodes.

The RM-ANOVA for the P3 amplitude revealed significant main effects of ***Time*** [F_(1,30)_ = 14.32, *p* = 0.001, *η_p_*^2^ = 0.32], ***Group*** [F_(1,30)_ = 58.08, *p* < 0.001, *η_p_*^2^ = 0.66], ***Condition*** [F_(1,30)_ = 45.12, *p* < 0.001, *η_p_*^2^ = 0.60], and ***Electrode*** [F_(1,30)_ = 543.94, *p* < 0.001, *η_p_*^2^ = 0.95], with the P3 amplitude being significantly greater in the post-test (1.79 ± 0.89 *μ*V) than in the pre-test (1.51 ± 0.40 *μ*V) across both groups, two conditions, and two electrodes, with the P3 amplitude being significantly greater in the EG (1.78 ± 0.32 *μ*V) than in the CG (1.25 ± 0.28 *μ*V) across two time points, two conditions, and two electrodes, with greater P3 amplitudes shown in the congruent condition (1.83 ± 0.69 μV) than in incongruent condition (1.48 ± 0.57 *μ*V) across the two groups, two time points, and two electrodes, and with greater P3 amplitudes being observed at the Cz site (2.24 ± 0.57 μV) than at the Fz site (0.79 ± 0.38 *μ*V) across the two groups, two time points, and two conditions. The interaction of ***Group*** × ***Time*** [F_(1,30)_ = 37.63, *p* < 0.001, *η_p_*^2^ = 0.63] was also significant. The *post-hoc* analyses revealed that the P3 amplitude was significantly larger at post-test as compared to pre-test in the EG (pre- vs. post-test: 1.78 ± 0.32 *μ*V vs. 2.51 ± 0.64 *μ*V, *p* < 0.001) across two conditions and two electrodes.

## 4. Discussion

To the best of our knowledge, this is the first study to investigate the effects of an acute intervention combining aerobic exercise and resistance exercise on behavioral and cognitive electrophysiological performance related to inhibitory control deficits in female adults with obesity. The main findings indicated that, although a single bout of acute aerobic-and-resistance exercise did not improve behavioral performance (e.g., ARs and RTs) in the obese women when performing the Stroop task, beneficial effects on the cognitive electrophysiological signals (e.g., shorter N2 and P3 latencies, smaller N2 amplitudes, and greater P3 amplitudes) were induced through this exercise mode intervention in the EG. In contrast, after a 30-min sitting rest, significantly slower N2 latency in the congruent condition was observed in the CG.

### 4.1. Behavior Performance

In the present study, acute aerobic-and-resistance exercise did not produce a significant behavioral (e.g., ARs and RTs) improvement in the Stroop test performance in the obese women, suggesting that the acute exercise intervention did not facilitate specific effects on the response inhibition/interference in this group. Previous studies have demonstrated that cognitive performance reflecting Stroop interference as measured by RTs improves significantly following an acute bout of either moderate-intensity aerobic exercise [12,52] or resistance exercise [53]. Quintero et al. [16] also found that acute aerobic HIIE and PRE + HIIE exercise interventions induce behavioral enhancements in cognitive inhibition in overweight inactive male adults when performing the Stroop test. In addition, overweight/obese female adults with type 2 diabetes showed reduced Stroop interference scores following an acute bout of moderate-intensity aerobic exercise [18]. The present results partly concurred with earlier studies investigating the effects of acute exercise on cognitive performance in older adults [54] or in obese groups [17,18]. For example, Vincent et al. [18] indicated that there was no significant effect of moderate-intensity acute aerobic exercise on Stroop performance among young overweight/obese adults or young adults with type II diabetes. Also, Tomporowski et al. [17] found that an acute bout of aerobic exercise did not improve error rates in overweight children when performing a task-switching task. In this study, no relationship was observed between acute exercise and behavioral performance in the obese participants. One plausible reason for the present finding could be that obese individuals with a sedentary lifestyle often exhibit worse VO_2max_ [7,35], and previous studies have suggested that cardiorespiratory fitness, but not acute exercise, could be an important factor modulating cognitive performance [55,56]. This conjecture is somewhat speculative, but provides a basis for future research.

### 4.2. Cognitive Electrophysiological Performances

An EEG is ideal for capturing the rapid brain neural processes involved in perceptual processing, which requires attention allocation for subsequent inhibitory success (e.g., ERP P2 component) [57,58,59] and inhibitory control (e.g., ERP N2 and P3 components) [50,51]. In spite of a lack of behavioral improvement in obese women induced by the acute aerobic-and-resistance exercise intervention when performing the Stroop task in the present study, beneficial effects on cognitive electrophysiological signals as measured by brain ERPs were observed in this study. Indeed, increased Stroop interference-related activation in the dorsolateral prefrontal cortices due to an acute bout of a combination of aerobic and resistance exercise was reported in healthy young adults [48]. Although the ERP P2 component is extremely sensitive to arousal levels [60,61], P2 latency and amplitude were not significantly improved after an exercise intervention in the EG in the present study. However, significant effects of acute moderate-intensity exercise on the modulation of the two inhibition-related ERP components (e.g., N2 and P3) were observed. The N2 component, the frontocentral early negative deflection occurring around 200–400 ms post-stimulus, is mainly related to early modality specific inhibition and conflict monitoring processes [49,50,51]. Drollette et al. [62] found smaller N2 amplitudes induced by acute exercise only for individuals with lower inhibitory capacity but not among those with higher capacity [63,64]. Increased adiposity has been linked with poorer inhibitory control abilities [4,5,7,35]. Although some studies examining the effects of acute exercise on executive function using a neurophysiological approach failed to observe alterations in N2 amplitude after acute aerobic exercise [9,55], the results of the present study showed that the decreased N2 amplitude following acute exercise in the EG was in line with the previous findings [62,65], implying that 30-min of supervised moderate-intensity aerobic dance combined with resistance exercise could enhance response inhibition associated with conflict monitoring [66] in obese women. Concurrently, the EG exhibited shorter N2 latencies after the exercise intervention as compared to the baseline whereas the CG exhibited significantly longer N2 latency in the congruent condition after a 30-min sit-and-rest. The findings suggested that acute exercise may produce more efficient neural processing involving the detection of a response inhibition process/conflict [67] in obese individuals.

The ERP P3 component, a positive component occurring around 300–800 ms post-stimulus, is typically associated with late general inhibition [50,51] and attentional resource allocation [68,69]. Several experimental studies have generally demonstrated increased amplitude and shortened latency of P3 components in relation to cognitive electrophysiological improvements caused by an acute bout of exercise [70,71,72]. Compatible with many reports of shorter P3 latency and larger P3 amplitude found after acute moderate-intensity aerobic exercise in healthy preadolescent children [62] and young adults [9,70] when performing the Flanker task and the Stroop task, respectively, similar improvements in cognitive electrophysiological signals were also observed following acute aerobic exercise combined with resistance exercise in the obese women in this study. The present findings suggest that faster cognitive processing to detect and process a stimulus in the environment [73] and more attentional resources allocated to process late general inhibition can be induced following acute exercise in obese individuals. Similarly, improvements in inhibition as assessed by the ERP P3 component of the cognitive tasks (e.g., task-switching, visuospatial attention task, and the Flanker task) following acute aerobic and resistance exercise interventions have also been reported in young [14,23,48] and older adults [10] as well as in individuals with developmental coordination disorders [9] and mild cognitive impairment [25], suggesting that acute exercise has a greater influence on cognitive functions involving diverse inhibitory control demands.

In terms of cognitive electrophysiological signals involved in inhibitory control, obese individuals have been demonstrated to show deviant ERP performance when performing the Stroop task [5]. Many empirical studies have reported that an acute bout of moderate-intensity aerobic exercise may elicit improved brain neuroelectric inhibition indices [9,10,23,25,48]. In addition, it has been observed that a single bout of acute resistance exercise effectively enhances cognition [74,75], but not many studies have so far examined the underlying electrophysiological processes, and the available findings are relatively heterogeneous [10,48]. According to the ACSM, moderate-intensity aerobic combined with resistance exercise for weight loss in overweight and obese adults is recommended for a minimum of 150 min each week to improve health [25]. However, among inactive and overweight/obese adults, increased exercise intensity appears to have a negative effect on affective responses [76]. Indeed, it has been reported that moderate-intensity exercise improves cognitive performance and increases prefrontal oxygenation to a greater extent [77]. Furthermore, it has been proposed that the relationship between acute exercise and cognitive performance follows an inverted U-shape [78]. However, it has to be acknowledged that this is not a universal finding since this relationship is influenced by several mediators (e.g., exercise intensity, time between exercise cessation, and cognitive testing) [79,80]. In the present study, we found that a 30-min bout of supervised moderate-intensity aerobic combined with resistance exercise program may have been effective in terms of improving cognitive interference inhibition in the obese female adults when they performed the Stroop task. Accordingly, the exercise prescription adopted in the present study seems to be feasible not only to lose weight but also to remedy the deficits of inhibitory control in such a group.

### 4.3. Limitations

The effect of exercise on the development of dietary obesity is different in males and females [81], and the adverse effects of obesity on neurocognitive functioning/performance are sex-specific [82]. Gender plays a significant role in pathophysiological changes and clinical manifestations due to a crucial effect of sex hormones on neurohumoral adipose tissue activity [83]. In addition, adiposity-related indices (e.g., body fat % and BMI) and low-grade systemic inflammation are considerably stronger in women than in men [84,85,86]. Weight reduction as a means to prevent a state of subclinical inflammation may be particularly effective in women [86]. Since inflammatory cytokines (e.g., TNF-α and C-reactive protein) were found to be significantly correlated with cognitive electrophysiological signals (e.g., ERP N2 and P3) in the obese group [7,35], an avenue for future research would be to examine the possibility of interactions among acute exercise, biochemical markers, and neurocognitive performance in both sexes. Additionally, although acute exercise can temporarily improve cognitive performance through arousal, the biochemical indicators that can actually promote the proliferation of brain neurons still cannot be effectively improved in a static state [23,25]. The most important way to improve neurocognitive performance and biochemical indices is to engage in long-term, regular physical exercise [24,26]. Also, a cross-sectional study indicated that, as compared to obese male adults with a sedentary lifestyle, obese individuals engaging in regular exercise can still obtain advantages with regard to neurocognitive performance [35]. Accordingly, further long-term regular exercise interventional studies are needed to understand the neurocognitive benefits for obese sedentary women.

## 5. Conclusions

This is the first study to investigate the effects of an acute exercise modality combining aerobic and resistance exercise on neurophysiological (i.e., behavioral and cognitive electrophysiological) performance among obese women. We found that brain neural processing related to early and late inhibition (e.g., ERP N2 and P3) were improved by the proposed exercise mode in obese female adults although behavioral benefits were not observed. Given that Stroop task performance may be related to cognitive interference inhibition, the present findings imply that the executive control networks and the efficiency of inhibitory control seem to remediable through an exercise intervention in obese individuals. However, the beneficial effects derived from the present findings through acute exercise and neurocognitive performance are temporary. Regular exercise has been demonstrated to effectively reduce the basal levels of inflammatory cytokines and compromised neural activity in obese individuals [35]. Further research is recommended to determine the long-term effects of a combination of aerobic and resistance exercise on inhibitory control, as well as to advance our present understanding of the real mechanisms and substantial benefits of the exercise–cognition relationships on the neurocognitive problems in individuals with obesity in the clinical setting.

## Figures and Tables

**Figure 1 brainsci-10-00767-f001:**
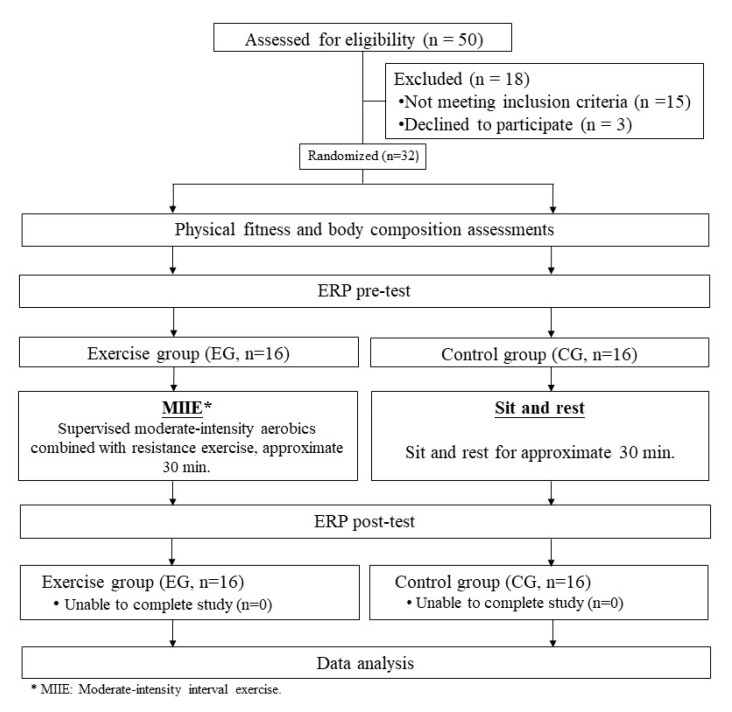
Schematic representation of experimental procedure.

**Figure 2 brainsci-10-00767-f002:**
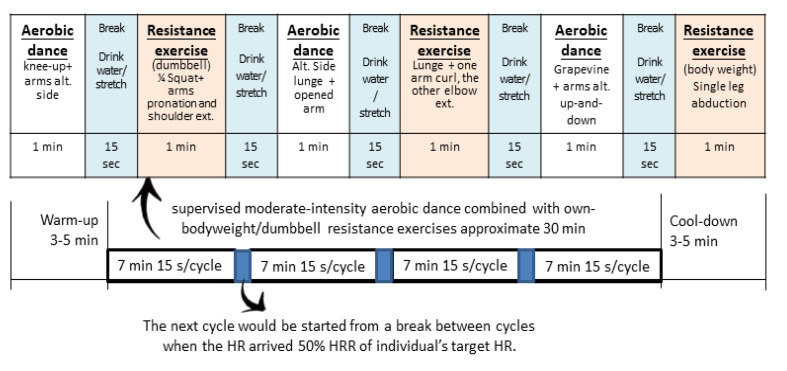
A single bout of exercise intervention.

**Figure 3 brainsci-10-00767-f003:**
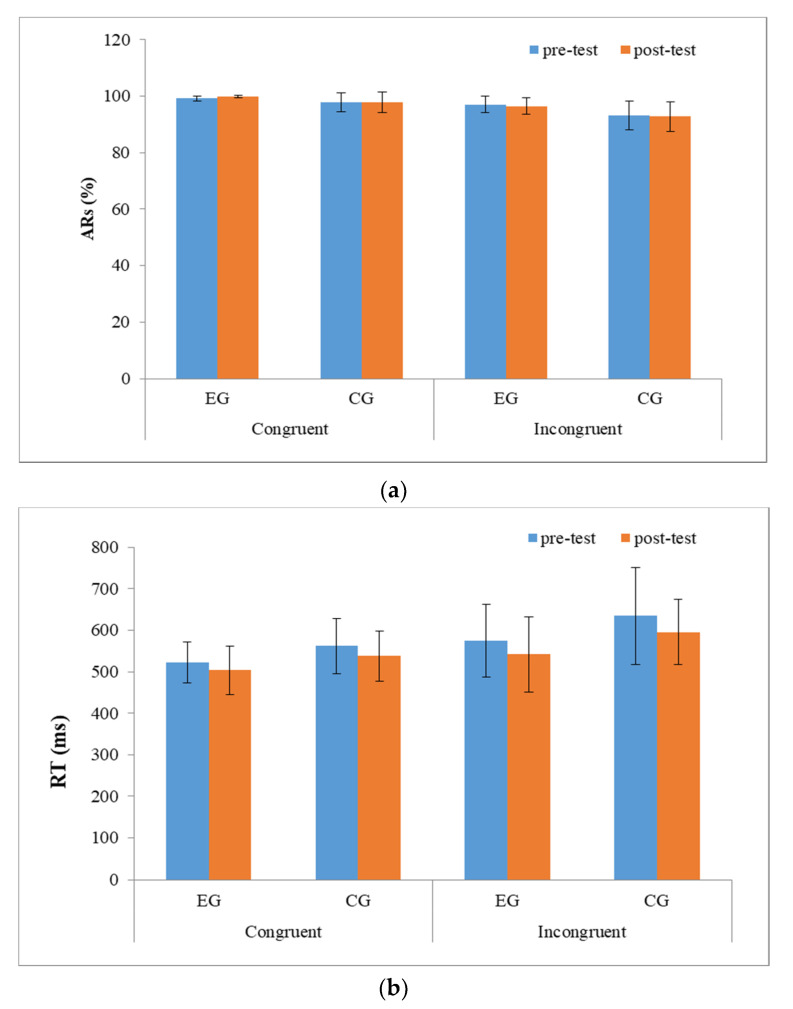
Behavioral performance (**a**) accuracy rates (ARs) (%) and (**b**) reaction times (RTs, ms) on the Stroop task in the exercise group (EG) and control group (CG) pre- and post-test.

**Figure 4 brainsci-10-00767-f004:**
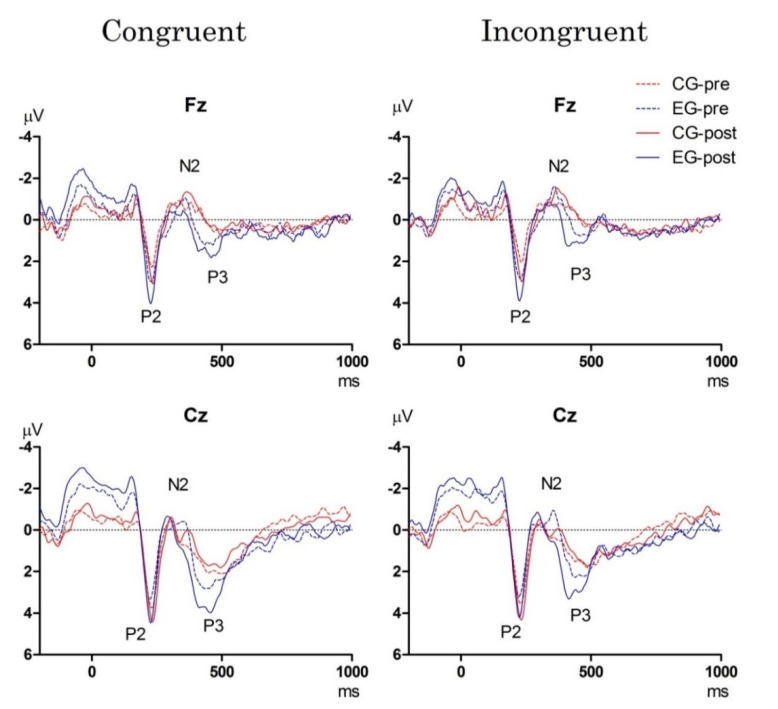
Grand averaged event-related potentials (ERPs) of P2, N2, and P3 waveforms in the congruent and incongruent conditions in two electrodes (Fz and Cz) for the exercise group (EG) and control group (CG) pre- and post-test when performing the Stroop task.

**Table 1 brainsci-10-00767-t001:** Demographic characteristics of the obese participants.

Characteristics	Exercise Group (*n* = 16)	Control Group (*n* = 16)	*t*	*p* Value
Age (years)	33.13 ± 6.27	32.92 ± 7.17	0.09	0.930
Height (cm)	160.72 ± 4.21	159.21 ± 5.69	0.86	0.400
Weight (kg)	79.82 ± 11.57	79.14 ± 18.10	0.13	0.900
BMI (kg/m^2^)	30.83 ± 3.61	31.07 ± 6.08	−0.14	0.891
SBP (mmHg)	116.73 ± 11.37	110.19 ± 14.46	1.40	0.174
DBP (mmHg)	77.20 ± 8.11	73.50 ± 9.99	1.13	0.269
Resting HR (bpm)	74.53 ± 8.67	75.69 ± 6.03	−0.43	0.668
Estimated VO_2_max	24.53 ± 4.88	24.66 ± 4.74	−0.07	0.943
Strength of leg extension (kg)	33.21 ± 9.74	35.81 ± 8.65	−0.79	0.438
BDI-II	6.25 ± 2.83	6.88 ± 2.94	−0.61	0.545
MMSE	29.44 ± 0.89	29.13 ± 0.83	0.98	0.336
PA energy expenditure (MET/day)	36.02 ± 13.18	32.34 ± 7.74	0.96	0.356
PA energy expenditure (kcal/day)	2522.19 ± 695.04	2652.49 ± 706.39	−0.49	0.631
Dietary (kcal/day)	2081.59 ± 579.98	1905.10 ± 558.73	0.82	0.420
Circumference				
Waist (cm)	91.81 ± 11.28	92.52 ± 17.19	−0.14	0.893
Abdominal (cm)	100.35 ± 11.85	102.84 ± 18.45	−0.45	0.660
Hip (cm)	113.64 ± 6.75	113.14 ± 14.59	0.12	0.905
Waist-Hip Ratio	0.81 ± 0.06	0.81 ± 0.06	−0.40	0.693
Percentage fat				
Whole body (%)	41.43 ± 3.89	44.12 ± 4.34	−1.85	0.075
Upper limbs (%)	48.71 ± 5.51	45.56 ± 3.92	1.86	0.073
Trunk (%)	42.23 ± 4.51	45.24 ± 5.38	−1.72	0.097
Lower limb (%)	41.40 ± 4.54	42.12 ± 4.25	−0.46	0.649

BMI, body mass index; SBP: systolic blood pressure; DBP: diastolic blood pressure; HR: heart rate; bpm: beat per minute; BDI, Beck Depression Inventory; MMSE, Mini-Mental State Examination; and PA, physical activity. Values are means ± SD.

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
