# Peer review of "Effects of Acute Aerobic Exercise Combined with Resistance Exercise on Neurocognitive Performance in Obese Women"

_brainsci, 2020, doi:10.3390/brainsci10110767_

Round 1
Reviewer 1 Report
The article has been significantly improved.
It has one few minor drawbacks:
1. Figure 1 is not a flowchart but an ordinary diagram. Flowcharts are not just rectangles. You should change the description below the picture and then in the article section so as not to refer to the flowchart.
2. In connection with the topic of EEG, I propose in the literature to quote one of the latest publications in this field, because most of the publications included in the study are not the latest - from 2020. For example, it could be a chapter from a book from 2020: Analysis and Classification of EEG Signals for Brain – Computer Interfaces.
3. Conslusions may be more saturated with plans for the future. The current description is general.
Author Response
Dear Editor-in-Chief and Assistant Editor Ms. Daisy Gao,
We are grateful for you and the two reviewers for providing some constructive suggestions to improve this manuscript again. It is hoped that the following responses and changes to the text adequately respond to the reviewers’ concerns.
Responses to Reviewer 1’s comments
The article has been significantly improved.
It has one few minor drawbacks:
- Figure 1 is not a flowchart but an ordinary diagram. Flowcharts are not just rectangles. You should change the description below the picture and then in the article section so as not to refer to the flowchart.
Reply: The Figure 1 legend was changed as the Reviewer #1’s suggestion. In addition, the description in the article has been modified. (Please see line 118 and line 120)
- In connection with the topic of EEG, I propose in the literature to quote one of the latest publications in this field, because most of the publications included in the study are not the latest - from 2020. For example, it could be a chapter from a book from 2020: Analysis and Classification of EEG Signals for Brain – Computer Interfaces.
Reply: The reference has been updated and added. (Please see Line 68 & the 20th reference (Lines 533-534)
- Conslusions may be more saturated with plans for the future. The current description is general.
Reply: The plans for the future have been added based on the Reviewer #1’s suggestion. (Please see Lines 464-465, and lines 469-471)

Reviewer 2 Report
The authors addressed all my comments in the revised version of their manuscript which significantly improved the overall quality of their article. But suggest to consider the following three minor comments on wording before publication.
- Line 108 to 110 – The authors wrote “Based on the previous findings mentioned above, we hypothesized that an acute bout of a program combining aerobic exercise and resistance exercise may be a feasible and effective means by which to improve cognitive deficits in obese female individuals.” I think the sentence could be revised as follows: “Based on the previous findings mentioned above, we hypothesized that an acute bout of a program combining aerobic exercise and resistance exercise is a feasible and effective intervention that can improve cognitive deficits in obese female individuals.”
- Line 126/ 127 – Can you please add the cut-off value used for the Edinburgh Handed Inventory (e.g., 50 as based on “Dragovic M. Categorization and validation of handedness using latent class analysis. Acta Neuropsychiatr. 2004;16:212–8. doi:10.1111/j.0924-2708.2004.00087.x.”)?
- Line 181/182 - I very thankful that the authors incorporate the important information about repetition velocity by adding the following to the text: “The repetition velocity of each resistance training movement (arm pronation and shoulder extension, arm curl, and elbow extension) was set at 8.4s to 2.1s.” However, in my opinion, it would be more clear to the reader what was done if the authors modified their sentence as follows: “The repetition velocity of each resistance training movement was set to: arm pronation (xx seconds), shoulder extension (xx seconds), arm curl (xx seconds), and elbow extension (xx seconds).”
Author Response
Dear Editor-in-Chief and Assistant Editor Ms. Daisy Gao,
We are grateful for you and the two reviewers for providing some constructive suggestions to improve this manuscript again. It is hoped that the following responses and changes to the text adequately respond to the reviewers’ concerns.
Responses to Reviewer 2’s comments
The authors addressed all my comments in the revised version of their manuscript which significantly improved the overall quality of their article. But suggest to consider the following three minor comments on wording before publication.
Line 108 to 110 – The authors wrote “Based on the previous findings mentioned above, we hypothesized that an acute bout of a program combining aerobic exercise and resistance exercise may be a feasible and effective means by which to improve cognitive deficits in obese female individuals.” I think the sentence could be revised as follows: “Based on the previous findings mentioned above, we hypothesized that an acute bout of a program combining aerobic exercise and resistance exercise is a feasible and effective intervention that can improve cognitive deficits in obese female individuals.”
Reply: The sentences have been modified according to reviewer #2’s suggestion. (Please see Line 109-110)
- Line 126/ 127 – Can you please add the cut-off value used for the Edinburgh Handed Inventory (e.g., 50 as based on “Dragovic M. Categorization and validation of handedness using latent class analysis. Acta Neuropsychiatr. 2004;16:212–8. doi:10.1111/j.0924-2708.2004.00087.x.”)?
Reply: The cut-off value was added. (Please see Line 127). Also, the reference was provided in the Reference section (Please see the 39th reference)
- Line 181/182 - I very thankful that the authors incorporate the important information about repetition velocity by adding the following to the text: “The repetition velocity of each resistancetraining movement (arm pronation and shoulder extension, arm curl, and elbow extension) was set at 8.4s to 2.1s.” However, in my opinion, it would be more clear to the reader what was done if the authors modified their sentence as follows: “The repetition velocity of each resistance training movement was set to: arm pronation (xx seconds), shoulder extension (xx seconds), arm curl (xx seconds), and elbow extension (xx seconds).”
Reply: Thank Reviewer #2 for the revised recommendation. The sentence was modified. (Please see Lines 183-185 and Figure 2)

Reviewer 3 Report
Dear Authors,
I have reviewed your manuscript again, even though there are no new experiments included in the manuscript. However, the manuscript text has significantly improved and clearer.
Author Response
Dear Editor-in-Chief and Assistant Editor Ms. Daisy Gao,
We are grateful for you and the two reviewers for providing some constructive suggestions to improve this manuscript again. It is hoped that the following responses and changes to the text adequately respond to the reviewers’ concerns.
Response to the Reviewer #3’s comment
Dear Authors,
I have reviewed your manuscript again, even though there are no new experiments included in the manuscript. However, the manuscript text has significantly improved and clearer.
Reply: Thank Reviewer #3 for the positive comment on our efforts in the revised manuscript.
This manuscript is a resubmission of an earlier submission. The following is a list of the peer review reports and author responses from that submission.
Round 1
Reviewer 1 Report
There is no doubt that none of studies have yet been conducted on the effects of a combination of acute aerobic and resistance exercise exercise on neurocognitive functions in obese individuals. As written in the article the aim of this study was thus to examine the effect of a single bout of such an exercise mode on neurocognitive performance in healthy sedentary obese women.
My comments to the article:
- I propose to divide the introduction into sections to make it clear
- the article requires correction in terms of text formatting, e.g. not all text is properly justified.
Reviewer 2 Report
The authors of „The Effect of Acute Aerobic Exercise Combined with Resistance Training on the Neurocognitive Performance of Obese Women” wrote an interesting an relevant article dealing with the effects of an acute intervention which combines aerobic and resistance exercises on cognition and electrophysiological parameters. I have some comments which can help the authors to improve the overall quality of their manuscript. In the following I will describe my concerns in more detail.
- Line 11 and 15 – Please remove the redundant terms (i.e., “correspondence” and “exercise”)
- Line 3, Line 21, Line 28, Line 460 – The authors use the term “training” and “exercise” relative synonymously throughout their manuscript. Unfortunately, “training” and “ exercise” refer to different constructs (see [1–4]). I recommend to use the term “exercise” instead of the term “training”. Please edited the mentioned statements.
- Line 18 – Please replace “divided into” by “assigned to”.
- Line 22 – I suggest to write “After the acute bout of combined aerobic and resistance exercise, … were assessed within one week.”
- Line 25, Line 123, Line 344, Line 406, Line 416, Line 422 - What do the authors mean by “cognitive electrophysiological performance”? This term is, in my opinion, relative uncommon. Please consider to use one of the following termini “electrophysiological signals, electrophysiological parameters, electrophysiological markers, neuroelectric indices”.
- Line 38 – The authors stated the following in their manuscript “In particular, attentional inhibition and inhibitory control have been demonstrated as cognitive processing problems in obesity…”. From my point of view, a more appropriate wording would be: “In particular, attentional inhibition and inhibitory control are cognitive domains that are affected negatively by obesity…”
- Line 4, Line 75, Line 102 – Please add “physical” prior to exercise.
- Line 43 – Please add a s to “individual”. It should read as follows: “individuals”
- Line 44 – The authors should check whether they refer to the appropriate construct as it is important to differentiate between “exercise”, “physical activity” and “fitness” [1–5]. Furthermore, they should indicate to which fitness dimension they refer (muscular fitness, cardiorespiratory fitness, motor fitness).
- Line 44 – In my opinion the sentence could be started as follows: “In this regard, cardiorespiratory fitness…”
- Line 45 – I suggest to change the sentence as follows: “Furthermore, higher cardiorespiratory fitness…”.
- Line 44 t0 47 – The authors did provide evidence for structural brain changes. As they measure functional brain changes via EEG they should add some literature on the effects of, for instance, physical fitness on functional level (e.g., ERPs). The authors could consider, for instance, the following literature [6–10]
- Line 78 – Please add “physically” prior to “active”. It should read as follows: “…physically active older adults…”
- Line 91 – I am not sure if “neurophysiological (e.g., ERPs) performance” is the appropriate term. Are ERPs an “performance” or rather an “process”, “parameters” or “marker”? Please edit the wording. In my opinion, it could read as follows: “…neurophysiological markers (e.g., ERPs)…”.
- Line 115 – The authors might consider to add following review [11] to the references to support their statement that “Some studies have proven that neurocognitive performance can be enhanced via acute and chronic resistance exercise [11]” as the mentioned review summarize neurocognitive changes in response to resistance exercises and resistance training.
- The methods section could be enriched by a Flowchart as recommended in CONSORT-Statement [12]. In this regard, the authors should also acknowledge and add to their manuscript (i) which procedure was used to randomize the participants (e.g., which software), (ii) how the allocation concealment was conducted, and (iii) if there is some blinding in there study (e.g., assessor, statistician). Moreover, the authors should indicate which study design was used by take reference to the classification scheme provided in Figure 6 in following publication [13].
- Line 134 to 136 – The authors should provide more details on the Power calculation (e.g., which software was used – G-Power?, which preplanned statistical analysis was considered – RM-ANOVA?). Furthermore, they should add a reference supporting their assumption of a moderate effect size. Moreover, the authors should carefully check their references as [46] did not refer to power calculation and [47] did not refer to a handedness inventory.
- Line 147 – With regard to the handedness inventory, the mean an SD as well as the used cut-off values should be provided in Table 1.
- Line 152 – Please add where the “cognitive neurophysiology laboratory” is situated (e.g., at the Tzu Chi University?)
- Line 167 and 168 – The authors stated that “Once the participants’ HR had returned to within 10% of pre-exercise levels, they completed the cognitive task along with ERP recording again.” Could the authors please add how long it takes on average that the HR returned to within 10% of pre-exercise levels? (by providing mean ± SD)
- Line 171 to 173 – Why did the authors perform the cardiorespiratory fitness assessment after the acute exercise session? In general, the cardiorespiratory fitness is assessed prior to the acute exercise session to determine the exercise intensity.
- Line 175 and following – The authors should provide the exact formula which was used to calculate the exercise intensity (e.g., HRR). Furthermore, they should acknowledge the drawbacks of this procedure to determine exercise intensity (e.g., by considering following literature [14,15]).
- Line 188 – The authors stated that the exercise intensity in resistance exercise was determined by 1RM. Unfortunately, they did not describe how and when the RM test was performed. Please add this information in the manuscript.
- In this regard, the authors should also add more details on exercise variables of the resistance exercises to characterize the intervention more appropriately. They should add information about (i) inter-set rest period, (ii) inter-exercise rest period, (iii) repetition velocity, (iv) muscle action, (v) range of motion, and (vi) if the exercise was performed to “volitional muscle failure” [11]. In this context, I suggest to add in Table 1 an overview on load (e.g., mean and SD) as done in their previous well-written publications [16,17]. If available, they should also add information on the internal load (e.g., relative perceived exertion [RPE] during the exercises).
- Line 191 – The authors should add the unit after “60.05 ±57 %”. Percentage of what?
- Line 233 – Why did the authors only analyze the electrodes Fz and Cz? This procedure should be appropriately justified and supported by references.
- Line 255 to 259 – The comparison of baseline values is not strictly necessary. The authors could/should check out following article [18].
- Line 324 and Line 336 – Please add a hyphen between “post” and “hoc” (post-hoc).
- Line 337 – Please add “at” after “larger”.
- Line 340/341 – I recommend to change “effects of acute aerobic exercise combined with a resistance exercise intervention on executive functions…” into “effects of an acute intervention combing aerobic exercises and resistance exercises on executive functions and electrophysiological parameters…”
- Line 346 – I would suggest to change “after a 30-min rest break” into “ after a 30-min sitting rest”
- Line 373 and following – The sentence starting with “In spite of…” is relative long and hard to read. Please try to simplify this sentence to make it more readable.
- Line 377 – The authors refer to the “dorsolateral prefrontal cortex”. How does that fit to the electrodes which were located on Fz and Cz?
- Line 384 to 388 – In the first sentence of this paragraph the authors refer to functional brain processes (N2 amplitude) whereas in the second sentence the authors refer to structural brain changes (grey matter volume). As there is, to the best of my knowledge, no direct link between these parameters, the authors should use a more carefully wording (e.g., no “indeed”).
- Line 389 to 391 – The sentence is hard to read and should be edited to make it more readable.
- Line 419 – I suggest to change “cognitive inhibitory” to “inhibition”.
- Line 420 – I recommend to edited the sentence as follows: “In addition, it has been observed that an single bout of acute resistance exercise can effectively enhance cognition, but not many studies has so far examined the underlying electrophysiological processes and the available findings are relative heterogenous [REF].”
- Line 423 – The authors wrote “As mentioned above,….” Unfortunately, the ACSM was not mentioned in the previous sentence. Hence, I suggest to edit the wording. In this context, the authors might consider recent literature [19].
- Line 426 – The authors refer to ventilatory thresholds. To which threshold they refer – VT1 or VT2? This important information should be added.
- Line 427 to 429 – I suggest to change the first part of the sentence into “In the literature, it is reported that moderate-intensity aerobic exercise improves cognitive performance and increases prefrontal oxygenation to a greater extent as high load resistance exercise and high-intensity aerobic exercise [REF]. Furthermore, it has been proposed that the relationship between acute exercise and cognitive performance follows and inverted U-shape [REF]. However, it has to be acknowledged that this is no universal finding and that this relationship is influenced by several mediators (e.g., exercise intensity, time between exercise cessation and cognitive testing)”.
- Line 432 and following – The authors stated that “Moderate-intensity aerobic exercise combined with low-intensity resistance exercise has been reported as the most effective exercise mode in terms of enhancing cognitive function via increased oxy-Hb concentration in the prefrontal cortex that activates nerves [16, 82] in individuals when performing the Stroop task 435 [82].” To the best of my knowledge, the quoted studies used to support this sentence did not deal with “moderate-intensity aerobic exercise combined with low-intensity resistance exercise”. Hence, I recommend to delete or carefully rewrite this sentence.
- Line 449-451 – What did the authors mean by “neuronal hyperplasia”? Did the authors refer to “neuroplasticity”? Furthermore, the sentence is hard to read and understand. Hence, I recommend that the authors revise this statement carefully in order to enhance its readability.
- Line 452 – The authors wrote that “The most important way to improve …” I think it is preferable to replace “The most important way...” by “An important way…”
References
- Herold, F.; Müller, P.; Gronwald, T.; Müller, N.G. Dose-response matters! – A perspective on the exercise prescription in exercise-cognition research. Front. Psychol. 2019, doi:10.3389/fpsyg.2019.02338.
- Herold, F.; Törpel, A.; Hamacher, D.; Budde, H.; Gronwald, T. A Discussion on Different Approaches for Prescribing Physical Interventions – Four Roads Lead to Rome, but Which One Should We Choose? J. Pers. Med. 2020, 10, 55, doi:10.3390/jpm10030055.
- Gronwald, T.; Budde, H. Commentary: Physical Exercise as Personalized Medicine for Dementia Prevention? Front. Physiol. 2019, 10, 726, doi:10.3389/fphys.2019.01358.
- Budde, H.; Schwarz, R.; Velasques, B.; Ribeiro, P.; Holzweg, M.; Machado, S.; Brazaitis, M.; Staack, F.; Wegner, M. The need for differentiating between exercise, physical activity, and training. Autoimmun. Rev. 2016, 15, 110–111, doi:10.1016/j.autrev.2015.09.004.
- Caspersen, C.J.; Powell, K.E.; Christenson, G.M. Physical activity, exercise, and physical fitness: Definitions and distinctions for health-related research. Public Health Rep. 1985, 100, 126–131.
- Hillman, C.H.; WEISS, E.P.; Hagberg, J.M.; Hatfield, B.D. The relationship of age and cardiovascular fitness to cognitive and motor processes. Psychophysiol. 2002, 39, 303–312, doi:10.1017/S0048577201393058.
- Scisco, J.L.; Leynes, P.A.; Kang, J. Cardiovascular fitness and executive control during task-switching: An ERP study. Int. J. Psychophysiol. 2008, 69, 52–60, doi:10.1016/j.ijpsycho.2008.02.009.
- Wang, C.-H.; Shih, C.-M.; Tsai, C.-L. The Relation Between Aerobic Fitness and Cognitive Performance. Journal of Psychophysiology 2016, 30, 102–113, doi:10.1027/0269-8803/a000159.
- Chang, Y.-K.; Chu, C.-H.; Wang, C.-C.; Song, T.-F.; Wei, G.-X. Effect of acute exercise and cardiovascular fitness on cognitive function: an event-related cortical desynchronization study. Psychophysiology 2015, 52, 342–351, doi:10.1111/psyp.12364.
- Pontifex, M.B.; Hillman, C.H.; Polich, J. Age, physical fitness, and attention: P3a and P3b. Psychophysiology 2009, 46, 379–387, doi:10.1111/j.1469-8986.2008.00782.x.
- Herold, F.; Törpel, A.; Schega, L.; Müller, N.G. Functional and/or structural brain changes in response to resistance exercises and resistance training lead to cognitive improvements – a systematic review. Eur Rev Aging Phys Act 2019, 16, 1676, doi:10.1186/s11556-019-0217-2.
- Schulz, K.F.; Altman, D.G.; Moher, D. CONSORT 2010 Statement: updated guidelines for reporting parallel group randomised trials. Trials 2010, 11, 32, doi:10.1186/1745-6215-11-32.
- Pontifex, M.B.; McGowan, A.L.; Chandler, M.C.; Gwizdala, K.L.; Parks, A.C.; Fenn, K.; Kamijo, K. A primer on investigating the after effects of acute bouts of physical activity on cognition. Psychol Sport Exerc 2019, 40, 1–22, doi:10.1016/j.psychsport.2018.08.015.
- Herold, F.; Aye, N.; Lehmann, N.; Taubert, M.; Müller, N.G. The Contribution of Functional Magnetic Resonance Imaging to the Understanding of the Effects of Acute Physical Exercise on Cognition. Brain Sci. 2020, 10, 175, doi:10.3390/brainsci10030175.
- Jamnick, N.A.; Pettitt, R.W.; Granata, C.; Pyne, D.B.; Bishop, D.J. An Examination and Critique of Current Methods to Determine Exercise Intensity. Sports Med 2020, doi:10.1007/s40279-020-01322-8.
- Tsai, C.-L.; Wang, C.-H.; Pan, C.-Y.; Chen, F.-C. The effects of long-term resistance exercise on the relationship between neurocognitive performance and GH, IGF-1, and homocysteine levels in the elderly. Front. Behav. Neurosci. 2015, 9, 471, doi:10.3389/fnbeh.2015.00023.
- Chang, Y.-K.; Tsai, C.-L.; Huang, C.-C.; Wang, C.-C.; Chu, I.-H. Effects of acute resistance exercise on cognition in late middle-aged adults: general or specific cognitive improvement? J. Sci. Med. Sport 2014, 17, 51–55, doi:10.1016/j.jsams.2013.02.007.
- Boer, M.R. de; Waterlander, W.E.; Kuijper, L.D.J.; Steenhuis, I.H.M.; Twisk, J.W.R. Testing for baseline differences in randomized controlled trials: an unhealthy research behavior that is hard to eradicate. Int. J. Behav. Nutr. Phys. Act. 2015, 12, 4, doi:10.1186/s12966-015-0162-z.
- O'Donoghue, G.; Blake, C.; Cunningham, C.; Lennon, O.; Perrotta, C. What exercise prescription is optimal to improve body composition and cardiorespiratory fitness in adults living with obesity? A network meta-analysis. Obes. Rev. 2020, doi:10.1111/obr.13137.
Reviewer 3 Report
The presented research methodologies are not enough to claim the beneficial effects of the exercise in neurocognitive performance in obese adults. More cognitive and neurophysiological experiments are required to support the claim. Since the research involves the human subject, I believe further experiments may not be executed or performed. Based on the current form of the manuscript, I am not able to recommend it for the publication in the Journal of Brain Sciences.